# Anti-Invasive and Anti-Migratory Effects of Ononin on Human Osteosarcoma Cells by Limiting the MMP2/9 and EGFR-Erk1/2 Pathway

**DOI:** 10.3390/cancers15030758

**Published:** 2023-01-26

**Authors:** Guowei Gong, Kumar Ganesan, Qingping Xiong, Yuzhong Zheng

**Affiliations:** 1Department of Bioengineering, Zunyi Medical University, Zhuhai Campus, Zhuhai 519041, China; 2Guangdong Key Laboratory for Functional Substances in Medicinal Edible Resources and Healthcare Products, School of Life Sciences and Food Engineering, Hanshan Normal University, Chaozhou 521041, China; 3School of Chinese Medicine, LKS Faculty of Medicine, The University of Hong Kong, Hong Kong 999077, China; 4Jiangsu Key Laboratory of Regional Resource Exploitation and Medicinal Research, Huaiyin Institute of Technology, Huai’an 223003, China

**Keywords:** ononin, invasion, migration, osteosarcoma, inhibitory effects

## Abstract

**Simple Summary:**

Osteosarcoma is the most prevalent orthotopic bone tumor. Due to its high metastatic properties, it has become the leading cause of cancer death worldwide. At this time, there is no effective treatment for osteosarcoma. Hence, we aimed to investigate the efficacy of ononin on osteosarcoma cell migration, invasion, and the underlying mechanisms. The dose-dependent manners of ononin treatment increased the expression of apoptosis and inhibition of cell proliferation in MG-63 and U2OS osteosarcoma cell lines through the EGFR-Erk1/2 signaling pathways. For consistency, we used the MG-63-xenograft mice model to confirm the in vivo anti-tumorigenic and anti-migratory efficacy of ononin. These findings suggest that ononin could be a potentially effective agent against the metastasis of osteosarcoma.

**Abstract:**

Osteosarcoma is a common malignancy of the bone. Due to its high metastatic properties, osteosarcoma becomes the leading cause of cancer death worldwide. Ononin is an isoflavone glycoside known to have various pharmacological properties, including antioxidant and anti-inflammatory activities. In the present study, we aimed to investigate the efficacy of ononin on osteosarcoma cell migration, invasion, and the underlying mechanisms. The in vitro anti-tumorigenic and anti-migratory properties of ononin were determined by MTT, colony formation, invasion, and migration in MG-63 and U2OS osteosarcoma cell lines. The results were compared with the standard chemotherapeutic drug, doxorubicin (DOX), as a positive control. The dose-dependent manners of ononin treatment increased the expression of apoptosis and inhibition of cell proliferation through the EGFR-Erk1/2 signaling pathways. Additionally, ononin significantly inhibited the invasion and migration of human osteosarcoma cells. For consistency, we used the MG-63-xenograft mice model to confirm the in vivo anti-tumorigenic and anti-migratory efficacy of ononin by inhibiting the protein expressions of EGFR-Erk1/2 and MMP2/9. According to the histological study, ononin had no adverse effect on the liver and kidney. Overall, our findings suggested that ononin could be a potentially effective agent against the development and metastasis of osteosarcoma.

## 1. Introduction

Osteosarcoma is the most prevalent orthotopic bone tumor that primarily affects adolescents and young people. The development of malignant tumors frequently occurs in the metaphyseal region of long bones, which originates from mesenchymal tissue [1]. Chemotherapy is often used in combination with surgery for patients with osteosarcoma [1]. The patient’s survival rate has recently increased due to improvement in treatment methods [2]. Nevertheless, due to the aggressive nature of the disease and its rapid progression, the prognosis for patients with metastatic osteosarcoma remains poor [3]. Metastasis and relapse of the disease continue to be the main causes of death for patients with osteosarcoma, with a 5-year survival rate of less than 20% after the development of metastases [4,5]. Conversely, few evidence-based treatments have been reported to enhance survival rates. Metastasis is common, with over 85% of cases occurring in the lung [6]. Many patients with osteosarcoma die of recurrence and metastasis within a short time span due to the failure of chemotherapy [7].

There have been persistent efforts to enhance survival rates by identifying possible therapeutics through animal studies and early phase trials of innovative medications. However, these efforts have only marginally improved the survival rate for people with metastatic osteosarcoma [8]. Therefore, finding potent drugs for the treatment of osteosarcoma is essential. In tumor cells, invasion, adhesion, and migration are the key steps in the progression of tumor metastasis [9,10]. Invasion into surrounding tissues, filtration into the bloodstream, migration, and survival in distant organs are all signs of tumor cell metastasis [4,8]. It may be possible to develop novel strategies for the treatment of osteosarcoma by elucidating the regulatory mechanisms governing invasion, adhesion, and migration, and this may be done by employing potential therapeutic approaches to prevent metastasis. Various plant-derived chemicals have already shown potential efficacy against a variety of other cancer types. Given the extraordinary lack of progress perceived in human osteosarcoma clinical trials that continue to employ many combinations of cytotoxic chemotherapy, it is time we take a closer look at these targeted drugs and natural compounds.

Ononin (formononetin-7-O-β-D-glucoside) is an isoflavone glycoside chiefly present in soybeans, fruits, Astragali Radix, Smilax scobinicaulis, Ononis angustissima, Millettia nitida, and herbal plants. It has a wide range of biological effects, such as antioxidant, antidiabetic, anti-obesity, neuroprotective, cardioprotective, antiviral, and anti-inflammatory qualities [11,12,13,14,15,16]. Ononin exhibits dose- and time-dependent manners of anti-cancer effects, including the larynx [17], colon [18], lung [19], and breast [20]. In addition, ononin inhibited ER stress and apoptosis, likely activating the SIRT3 pathway, and protected against doxorubicin-induced cardiotoxicity [21]. In order to treat osteosarcoma, alternative medications with minimal or no side effects are highly needed. Hence, the present study aimed to investigate whether ononin inhibits osteosarcoma cell migration, invasion and proliferation through EGFR-Erk1/2 and MMP2/9 in MG-63, and U2OS and MG-63 xenograft mice.

## 2. Materials and Methods

### 2.1. Cell Culture

The MG-63 and U2OS cell lines were provided by the American Type Culture Collection (ATCC, Manassas, VA, USA). These cell lines were grown in Dulbecco’s modified Eagles medium, supplemented with 100 IU/mL penicillin, 100 g/mL streptomycin, and 10% fetal bovine serum. They were altered to produce the firefly luciferase gene. Cells were grown in a water-saturated 5% CO_2_ incubator at 37 °C.

### 2.2. MTT Assay

The MTT test was used to measure cell viability. The MTT assay kit was purchased from Sigma-Aldrich (St. Louis, MO, USA). In 96-well plates, cells were plated. Ononin was given from 0–3 μM. MTT solution was added to the cultures after 48 h of treatment at a final concentration of 0.5 mg/mL. The purple crystal growth was dissolved in DMSO after 2 h of incubation. For calibration, an absorbance of 570 nm was used. The percentage of absorbance of the blank group was used to calculate cell viability, and the value was set to 1.

### 2.3. Colony Formation Assay

One thousand cells were planted in a 6-well plate and treated with different concentrations of ononin for 48 h before being replaced with new media for another 7 days. The cells were fixed in methanol for 15 min at room temperature before being stained with crystal violet (Sigma-Aldrich) for 10 min.

### 2.4. Flow Cytometry Analysis

Prior to treatment, cells were seeded and cultured on 35 mm culture plates and allowed to grow for 24 h. Flow cytometry was used to label the cells with dyes, including Annexin V-FITC/propidium iodide (PI) or SYTOX AADvanced Fluorescence (Thermo Fisher Scientific, Grand Island, NY, USA). Cells were washed twice with phosphate-buffered saline before being collected in PBS. Cells were incubated for 15 min at room temperature in the dark in 100 μL of binding buffer. The samples were gathered automatically with the loader using a 10,000 event acquisition criterion for each tube. Quadrants were constructed based on the viable population of cells labeled with different dyes. The FlowJo v7.6 application was used to evaluate the results.

### 2.5. Cell Migration 

In brief, 1 × 10^5^ cells in 200 μL serum-free medium were incubated in the upper transwell chambers and treated with different doses of ononin treatments. The lower chambers were filled with 600 μL of 10% FBS medium. The non-migratory cells on the upper surface of the chambers were removed with a cotton swab after 48 h, and the migrated cells were fixed in 4% paraformaldehyde and stained with 0.2% crystal violet.

### 2.6. Cell Invasion

Cell invasion tests were conducted in 24-well transwells coated with Matrigel (Thermo Fisher Scientific). In brief, 1 ×10^5^ cells were cultured in 200 μL serum-free DMEM, and treated with various doses of ononin in the upper transwell chambers. An amount of 600 μL of 10% FBS medium were placed in the lower chambers. After 48 h, the cells on the chamber’s upper surface were scraped with a cotton swab. The invaded cells were fixed in 4% paraformaldehyde and stained with 0.2% crystal violet.

### 2.7. Western Blot

SDS-PAGE was used to determine protein expression in cell lysates. After transferring the target proteins to membranes, the membranes were treated with 1:1000 dilutions of anti-EGFR (CST, Danvers, MA, USA), anti-Ras (CST) at 1:1000 dilutions, anti-Raf (CST) at 1:1000 dilutions, anti-MEK (CST) at 1:1000 dilutions, anti-Erk1/2 (CST) at 1:1000 dilutions, anti-MMP-2 (CST) at 1:2000 dilutions, anti-MMP-9 (CST) at 1:2000 dilutions, anti-Vimentin (CST) at 1:2000 dilutions, anti-GAPDH (CST) at 1:1,000,000 dilutions, at 4 °C for overnight. The immunological complexes were detected by the enhanced chemiluminescence (ECL) method after 3 h of incubation with horseradish peroxidase (HRP)-conjugated secondary antibodies at room temperature (Amersham Biosciences, Piscataway, NJ, USA). Image J was used to compute and analyze the band intensities.

### 2.8. Animals

BALB/c nude mice (8–10 weeks old) were purchased from Shanghai Laboratory Animal Company (Shanghai, China). The animals were housed in a pathogen-free environment with a temperature of 22 °C ± 2 °C, relative humidity of 70%, and a 12 h light/dark cycle. Subcutaneously, MG-63 cells (2 × 10^8^ cells/mL) were injected into each mouse’s flank. Five days after the cell injection, palpable and measurable tumors were identified. The animals were then randomly assigned to one of three treatment groups: (i) a control group that received PBS; (ii) a DOX group that received DOX (2 mg/kg/i.p); and (iii) an ononin group that received a different dosage of ononin (1, 3, and 10 mg/kg/i.p). The experimental procedure was approved by Zunyi Medical University’s Animal Experimentation Ethics Committee (No. 21-029 for Animal Ethics Approval) and followed the “Principles of Laboratory Animal Care.” After 42 days of treatment, the mice were sacrificed, and tumor weights and sizes were recorded.

### 2.9. Bio-Illuminescence Assay in the Xenograft Model

The mice were anesthetized with isoflurane before being injected intraperitoneally with 100 μL of aminoluciferin (0.5 mM, diluted in PBS), followed by 3 min of bioluminescence imaging, then the animals were euthanized for further research. The collected lung tissue was also subjected to bioluminescence imaging. The IVIS^®^ Spectrum imaging system (PerkinElmer, Alameda, CA, USA) was used to collect photos, and the fluorescence was analyzed using Living Image software (PerkinElmer).

### 2.10. Histology Analysis

After collecting lung tissue, it was washed in PBS before being fixed with 4% polyformaldehyde, dehydrated with gradient ethanol, and embedded in paraffin wax. The paraffin slices were then stained, and prepared using a slicing machine.

### 2.11. Serum Biochemical Analysis

Following the manufacturer’s instructions, plasma was collected and an automatic biochemical analyzer (7080 type, Hitachi, Tokyo, Japan) was used to test ALT (Abcam, MA, USA) and AST (Abcam) activity.

### 2.12. Statistical Analyses 

Protein concentrations were calculated using Bio-Bradford Rad’s protein assay dye and BSA as standards (Bio-Rad Laboratories, Hercules, CA, USA). To compare the means of distinct cell groups, one-way ANOVA (Bonferroni’s post-test) was performed.

## 3. Results

### 3.1. Ononin Exhibits Antiproliferative Activity against Human Osteosarcoma MG-63 and U2OS Cells

Initially, we optimized the ideal DOX concentration both on MG-63 and U2OS cells before proceeding with the biological analysis. We used the MTT test to detect the optimal concentration of DOX in two osteosarcoma cell types (Figure 1A)**.** DOX suppressed the cell proliferation of MG-63 or U2OS cells in a dose-dependent manner. DOX at 1 μM was found to be minimally toxic to MG-63 or U2OS osteosarcoma cells in comparison to the control group. As a result, DOX at 1 μM was selected for the following experiments (Figure 1A)**.** On the other hand, we used the MTT assay to determine the different concentrations of ononin in MG-63 or U2OS cells (Figure 1B)**.** Ononin at 1 μM was able to induce cell death with statistical differences (Figure 1B). Furthermore, we performed a colony formation assay to verify the effect of ononin on both MG-63 and U2OS cell treatments (Figure 1C). Based on staining data, ononin reduced colony formation in a dose-dependent manner (Figure 1C)**.**

### 3.2. Ononin Promotes Apoptosis in Human Osteosarcoma MG-63 and U2OS Cells

In addition to the MTT and colony formation assays, the induction of cell apoptosis was studied in both MG-63 and U2OS cells treated with 1 μM DOX or different concentrations of ononin (Figure 2A,B). Flow cytometry was used to examine Annexin V-FITC- and PI-labeled MG-63 or U2OS cells, which exhibited apoptosis after DOX therapy. DOX at 1 μM showed a significantly increased apoptosis rate as compared to the control group on MG-63 or U2OS cells (Figure 2A,B). Furthermore, a succession of ononin treatments boosted the rate of cell death, as reflected by a flow cytometer. The data were congruent with caspase 3/7 results, which were dose-dependently increased after 48 h of ononin treatment as indicated by flow cytometry (Figure 2C,D).

### 3.3. Anti-Invasive and Anti-Migratory Effects of Ononin on Human Osteosarcoma MG-63 and U2OS Cells

We used a matrix gel-coated transwell test for detecting the anti-migration and anti-invasive capacity of ononin on MG-63 or U2OS osteosarcoma cells (Figure 3). After 48 h of incubating ononin with MG-63 and U2OS, the cell’s migratory function was suppressed dose-dependently (Figure 3A). The cell attachment to the bottom layer was significantly reduced following 1 μM ononin treatment (Figure 3A). Similarly, the invasion functions of osteosarcoma cells were significantly decreased after 48 h of treatment with ononin, which was determined by the Matrix gel-coated transwell test (Figure 3B). The crystal violet staining of invading cells was identified in the bottom layer shown in purple (Figure 3B)**.**

### 3.4. Ononin Inhibits the Expression of EGFR-Erk1/2 Signaling in Human Osteosarcoma MG-63 and U2OS Cells

Osteosarcoma cells express markedly distinct levels of functional EGFR, which is well-recognized by EGF-mediated downstream cascade activation of Ras, Raf, MEK, and Erk [22]. We used SDS-PAGE to investigate the precise mechanism of this bioactive compound for the treatment of osteosarcoma. Ononin suppressed the cascade expressions of EGFR, Ras, Raf, MEK, and Erk1/2 proteins in a dose dependent manner. Furthermore, the results demonstrated that ononin at 1 μM had the greatest effective concentration for blocking target gene expression on MG-63 or U2OS cells (Figure 4). Similarly, treatment with DOX at 1 μM (as a positive control) significantly inhibited the cascading mechanism of EGFR, Ras, Raf, MEK, and Erk1/2 expressions with significant differences when compared to the control (Figure 4).

### 3.5. Ononin Inhibits Metastasis through the Expression of MMP2/9 in Human Osteosarcoma MG-63 and U2OS Cells

In many cancers, MMP2/9 performs an active role in cell proliferation, invasion, and tumor metastasis. In osteosarcoma tissues, MMP-2/9 contributes to the invasion of endothelial cells by destroying extracellular collagen [23,24]. In the present study, we also found that the metastasis-related biomarkers MMP-2/9 and vimentin were significantly reduced after treatment with ononin. The blotting data showed that after 48 h of incubation with ononin on MG-63 and U2OS cells, the activities of MMP-2, MMP-9, and vimentin were significantly reduced (Figure 5). A series of concentrations of ononin blocked the expression levels of MMP-2, MMP-9, and vimentin at varied levels, as shown in Figure 5. The results of SDS-PAGE revealed that the most effective dose of ononin for the anti-metastatic effect was 1 μM.

### 3.6. The Anti-Proliferative and Anti-Metastatic Effect of Ononin on MG-63 Xenograft Mice

In order to establish the anti-cancer effects of this bioactive compound, we used MG-63 human osteosarcoma xenograft nude mouse models, which showed encouraging outcomes from the preliminary in vitro data (Figure 6). After 42 days of treatment, the tumor area identified by IVIS had dramatically shrunk when compared to the blank control (Figure 6A). During a 42-day treatment period, DOX reduced cell viability (Figure 6A). However, as presented in Figure 6B, this chemotherapeutic treatment significantly reduced body weight in nude mice in comparison to the ononin-treated groups. Notably, the treatment with ononin significantly decreased tumor size and weight when compared to the control group (Figure 6C–E).

The animals were sacrificed after receiving treatment for 42 days. Lung tissues were collected from each treatment group. The bioluminescence data indicated that ononin reduced MG-63 lung metastasis in a dose-dependent manner compared to the control group (Figure 7A). In vivo bioluminescence imaging has gained potential for the measurement of lung metastasis and is frequently used in preclinical studies. This bioluminescence imaging study described a correlation between signal intensity and tumor mass. The data further showed that the number of lung metastases was much higher in the control group than in the ononin treatment group (Figure 7A).

The lungs were fixed after autopsy, and the number of lung metastases were marked (red arrow) in Figure 7B. Additionally, lung metastasis was significantly larger in the control group than in the ononin or DOX treatment groups when the lung tissue was stained with H and E (Figure 7C). The aforementioned evidence suggested that ononin was effective at treating the proliferation and metastasis of osteosarcoma in vivo, thus, ononin might be a therapeutic candidate for osteosarcoma. However, further research is required to determine the precise underlying mechanism.

We also examined the expression of cell apoptosis biomarkers because ononin reduced tumor size and volume in MG-63-xenograft mice. The results of the flow cytometer revealed cell status and caspase 3/7 gene expression activities in the homogenized tumor tissue (Figure 8). Figure 8A,B showed that treatment with ononin increased the rate of tumor cell death in MG-63 cells compared to the control group. Quantitative analysis of data also showed that incubation with ononin increased both the early and late stages of apoptotic osteosarcoma when compared to the control group (Figure 8A,B). The results were in line with data on caspase 3/7 protein expression, which confirmed that ononin treatment reduced the activation of caspase-3/7 (Figure 8C,D).

In addition, we examined the gene expression levels of EGFR, Ras, Raf, MEK, and Erk1/2 in the ononin-treated group (Figure 9A). The dose-dependent effects of ononin reduced target gene expression according to Western blot analysis (Figure 9A). DOX also inhibited the activities of the EGFR-Ras-Raf protein in comparison to the control group (Figure 9A). Furthermore, Western blotting was used to examine the protein associated with anti-metastasis after receiving the ononin treatment. Treatment with increasing doses of ononin significantly reduced metastasis in MG-63 xenograft mice (Figure 9B). Furthermore, DOX treatment was also capable of blocking MMP-2, MMP-9, and vimentin expression (Figure 9B).

### 3.7. The Safety of Ononin in Nude Mice after 42 Days of Treatment

In this study, the effects of ononin on inhibiting osteosarcoma cell proliferation and metastasis have been demonstrated in vitro and in vivo. Specifically, it is essential to determine whether ononin is toxic in vivo or not. During the animal experiments, there were no signs of pain or toxicity after ononin treatment, and all of the experimental mice survived up to 42 days after treatment. The H and E staining technique was used to exhibit the various organs obtained from the various treatment groups (ononin and DOX) in Figure 10. There were no visible tissue lesions found in the histological specimens obtained from the ononin experimental groups (Figure 10). Likewise, there was no significant evidence of cardiac fibrosis, liver inflammation, and glomerular or tubular changes in the kidneys (Figure 10). However, mice treated with DOX showed significant morphological changes in the shape of their hearts, livers, and kidneys when compared to the control group (Figure 10). Furthermore, we determined the ALT and AST levels in each group of animals to provide more biosafety data after ononin treatment, and the results showed that the ALT and AST contents were not altered after 42 days of ononin treatment (Figure 11). DOX, on the other hand, was capable of increasing ALT and AST levels (Figure 11). Thus, based on in vitro and in vivo studies, ononin could be the safe agent for reducing the proliferation and metastasis of osteosarcoma.

## 4. Discussion

Osteosarcoma cells are primarily mesenchymal cells that derive from long bones and are the most common bone cancer globally [25]. Metastasis is the main cause of clinical manifestations; however, the underlying mechanisms involved in metastasis and chemoresistance in the development of osteosarcoma remain unclear. Ononin is an active principle mostly found in fruits and vegetables that possesses several biological activities including anticancer properties [26]. Currently, no comprehensive study has been conducted on ononin and osteosarcoma. Hence, this study sought to investigate whether ononin inhibits osteosarcoma cell progression and blocks invasion, and migration in human osteosarcoma MG-63 and U2OS cells, and MG-63-xenograft mice.

Averting the proliferation of tumor cells is a key approach to controlling cancer [27]. Accordingly, we studied the antiproliferative effect of ononin on human osteosarcoma MG-63 and U2OS cells by using both MTT and colony formation assays. The results showed that the dose-dependent manners of ononin significantly inhibited the proliferation of MG-63 and U2OS cells. Furthermore, flow cytometry analysis reported that the dose-dependent manners of ononin dramatically enhanced the apoptotic rate in human osteosarcoma cells and MG-63-xenograft mice. This study was consistent with an earlier report that chlorogenic acid, a polyphenol, induced significantly apoptosis and cell death in MG-63 and U2OS cells [28]. In this present study, we compared the effectiveness of DOX on osteosarcoma cells with that of ononin, as a positive control. DOX is an anthracycline presently permitted to treat numerous neoplastic disorders, such as breast and gynecologic cancers, lymphoma, and lung cancer [29]. Additionally, DOX-based chemotherapy has been recognized as the standard treatment for osteosarcoma [30]. 

DOX promotes tumor cell death in several ways. Notably, DOX disrupts both DNA synthesis and replication by affecting topoisomerase II activation and intercalating DNA base pair mechanisms [31]. Dose-dependent treatment with ononin, or DOX, significantly accelerates the rate of apoptosis, resulting in osteosarcoma cell death. Apoptosis is a type of programmed cell death that involves the degradation of cytoskeletal components and is caspase-dependent [32,33]. Hence, we investigated whether ononin was able to induce apoptosis using caspase 3/7 in MG-63 and U2OS cells. Results also indicated that ononin or DOX caused apoptotic induction in MG-63 and U2OS cells.

Cell invasion and migration are two key cellular processes for cancer metastasis from the primary tumor to distant sites. These processes usually involve the breakdown of the extracellular matrix (ECM) and the epithelial-mesenchymal transition. These breakdowns are normally facilitated by proteolytic enzymes, viz., matrix metalloproteinases (MMPs). In general, MMPs facilitate tumor invasion and migration toward the vascular or lymphatic systems, resulting in metastatic dissemination [34]. There are more than 20 MMPs identified, and MMP-2 and MMP-9 may perform a role in invasiveness, migration, and metastasis. Increased levels of MMP2/9 are connected to invasion, migration, and metastasis, which are related to poor prognosis in patients with cancer [35]. In the transwell assay, the effects of ononin or DOX on MG-63 and U2OS cell invasion and migration were dose-dependently suppressed after 48 h of incubation. Furthermore, we investigated the inhibitory role of invasion and migration using the level of MMP2/9 expression in MG-63 and U2OS cells. Results also indicated that ononin or DOX caused anti-invasive and anti-migratory functions in MG-63 and U2OS cells. This present study was consistent with reports from earlier studies that treatment with chalcones, rosmarinic acid, and resveratrol inhibited MMP2/9 expression, resulting in the anti-invasive and anti-migratory functions of osteosarcoma [34,35,36]

There is mounting evidence that EGFR/Ras/Raf/Erk1/2 cascade activation facilitates key tumor progressions, including proliferation, migration, metastasis, and angiogenesis, by regulating downstream pathways [37,38]. It is well-recognized that aberrant changes in this signal are connected to osteosarcoma development. Thus, we examined the expression levels of EGFR/Ras/Raf/Erk1/2 in ononin or DOX-treated cells. The present data reported that treatment with ononin or DOX blocked the EGFR/Ras/Raf/Erk1/2 cascade signaling pathway in MG-63 and U2OS cells. In both in vitro and in MG-63-induced xenograft mice, results were consistent in showing that combined treatment with ononin, or DOX blocked the EGFR-Erk1/2 cascade regulatory pathway. This present study was consistent with reports from earlier studies that treatment with *Polygonum cuspidatum*, curcumin, and butein inhibited the EGFR-Erk1/2 cascade signaling pathway, resulting in the inhibition of osteosarcoma [39,40,41]. The ononin did not adversely affect the liver and kidney in the histological and AST/ALT studies, which indicates that it is a safe molecule. 

Our study is the first to demonstrate that ononin has anti-osteosarcoma activity against MG-63 and U2OS cells, and MG-63- xenograft mice by limiting the expression of MMP2/9. This comparable anti-cancer impact was shown with the chemotherapeutic agent DOX in the xenograft model, and the outcome was no evident cytotoxicity. As a result, ononin is offered as a viable molecule for the development of an innovative anti-tumor medicine candidate for osteosarcoma.

## 5. Conclusions

Ononin, an isoflavone glycoside, inhibits osteosarcoma progression in MG-63 and U2OS cells and MG-63-xenograft mice. Specifically, the treatment of ononin induces apoptosis, inhibiting cell proliferation, invasion, migration, and metastasis via blocking MMP2/9 expression and EGFR-Erk1/2 pathways, which was further validated using biomarkers in human osteosarcoma MG-63 xenograft mice. Overall, our findings suggest that ononin has value for the development of a novel anti-tumor treatment for osteosarcoma without side effects. Furthermore, well-designed clinical trials are essential to validate our understanding of the pharmacological functions and anti-tumor treatment of ononin in osteosarcoma. 

## Figures and Tables

**Figure 1 cancers-15-00758-f001:**
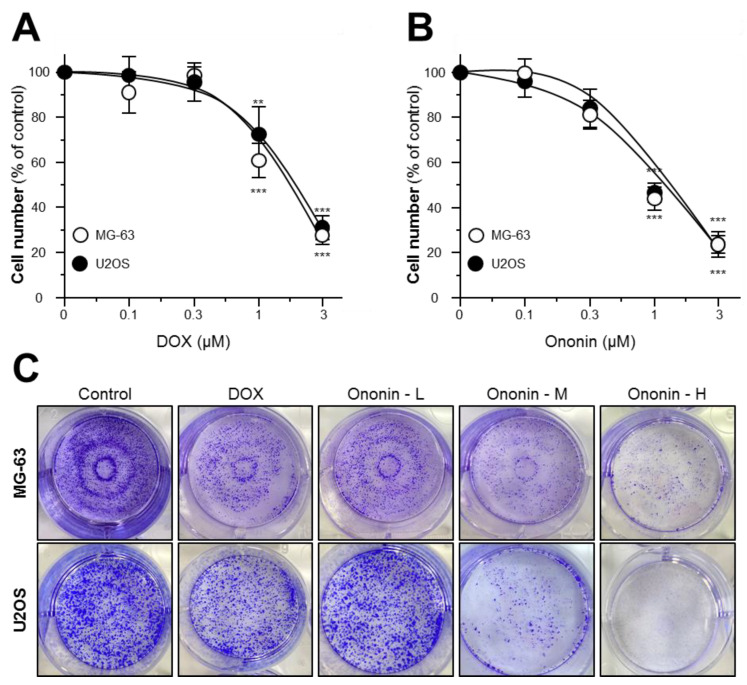
Cultured cells were treated with (**A**) DOX (0.1–3 μM) or (**B**) ononin (0.1–3 μM) for 48 h. The cell viability was determined by the MTT assay. Data are presented as a percentage change from the control group and in Mean ± SEM, with *n* = 6. Statistical changes were clustered as significant *** where *p* < 0.001 as compared with the control group. (**C**) Cells were seeded into 6-well plate, and after 48 h of treatment, cells were treated with fresh DMEM for another 7 days. Ononin at 0.1 μM was labeled as -L, 0.3 μM was highlighted as m and 1 μM was shown as -H, respectively. Crystal violet was used to stain the colony formation assay.

**Figure 2 cancers-15-00758-f002:**
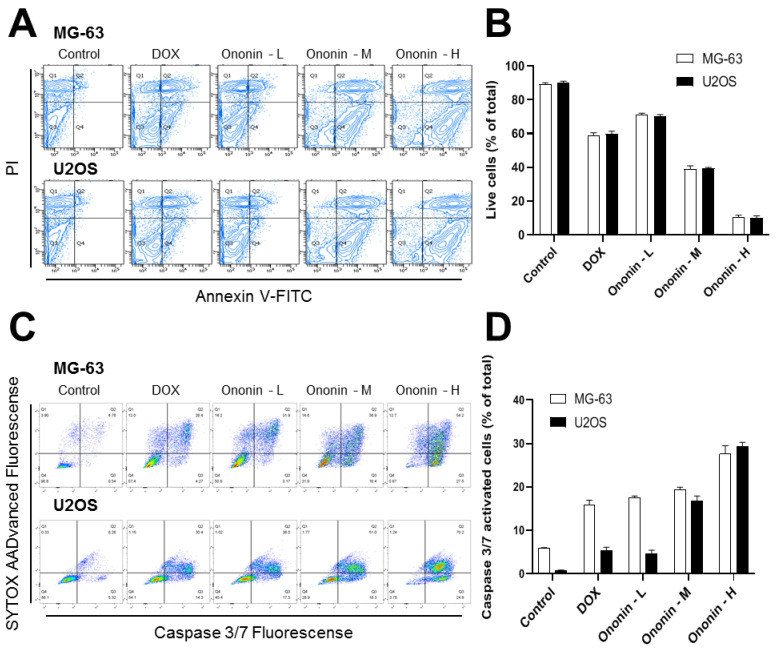
(**A**) The viable cell population was displayed in the bottom left quadrant, the early apoptotic cells in the bottom right quadrant, and the late apoptotic cells in the top right quadrant of the dual parametric dot plots integrating annexin V-FITC and PI fluorescence. (**B**) The quantification analysis was analyzed by the program Flowjo v7.6. (**C**) The dual parametric dot plots combining caspase 3/7 and SYTOX AADvanced fluorescence showed activated caspase 3/7 in the bottom right quadrant (Q4). (**D**) The quantification analysis was conducted by the program Flowjo v7.6. Ononin at 0.1 μM was labeled as -L, 0.3 μM was highlighted as -M, and 1 μM was shown as -H. The values are expressed as the percentage of total cell number, *n* = 6, in Mean ± SEM. DOX (1 μM) was applied as a positive control.

**Figure 3 cancers-15-00758-f003:**
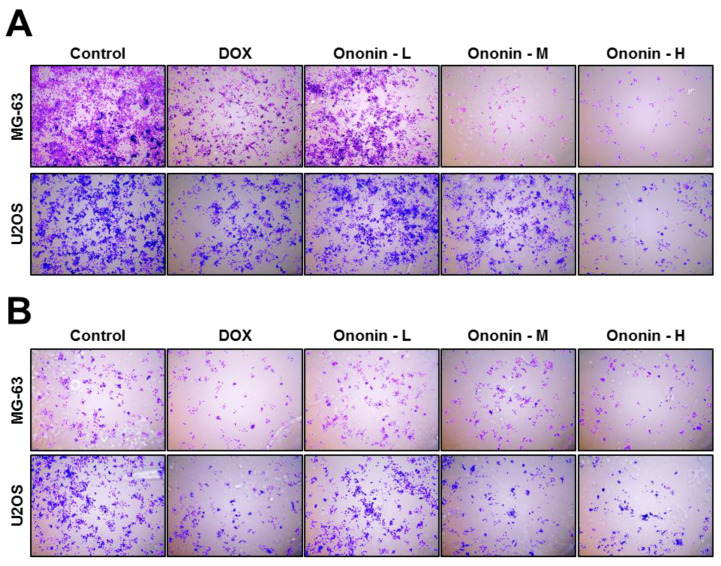
(**A**) Transwell assay analysis of MG-63 or U2OS treated with ononin for migration activity. (**B**) Matrigel-coated transwell was used here for evaluating the anti-invasion potential of ononin on osteosarcoma cells. Ononin at 0.1 μM was labeled as -L, 0.3 μM was highlighted as -M, and 1 μM was shown as -H.

**Figure 4 cancers-15-00758-f004:**
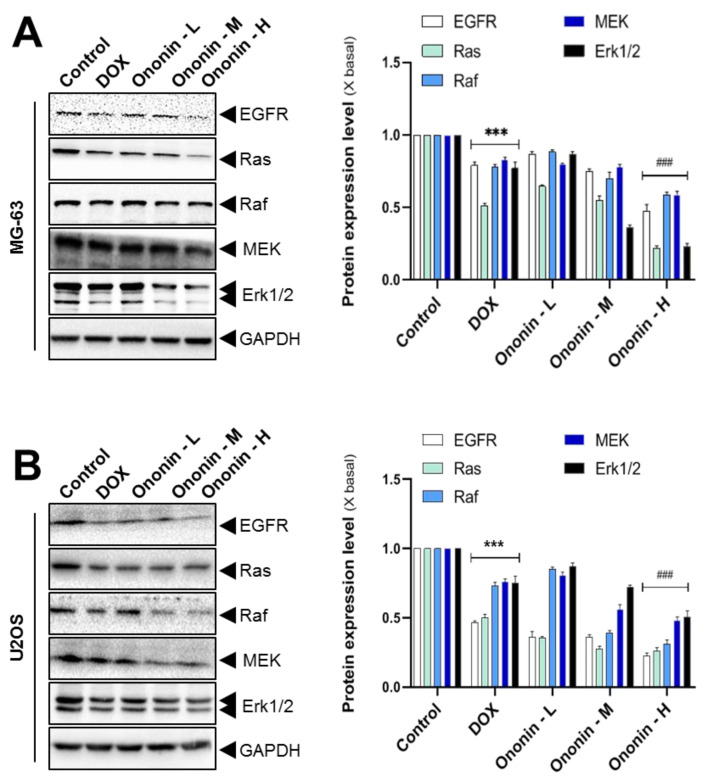
(**A**) Cultured MG-63 or (**B**) U2OS cells were incubated with DOX (1 μM) or ononin (0.1, 0.3, and 1 μM, labeled as -L, M, and –H) for 2 days, and the target proteins were detected by Western blot (left panel). The quantification of the target protein was calculated with a densitometer (right panel). The values are expressed as the fold of change (X basal), in Mean ± SEM, where *n* = 3. *** *p* < 0.01 was considered a significant result when compared to control, and ^###^ *p* < 0.001 was considered a significant result when compared to DOX group. Full Western blot images can be found at Appendix A.

**Figure 5 cancers-15-00758-f005:**
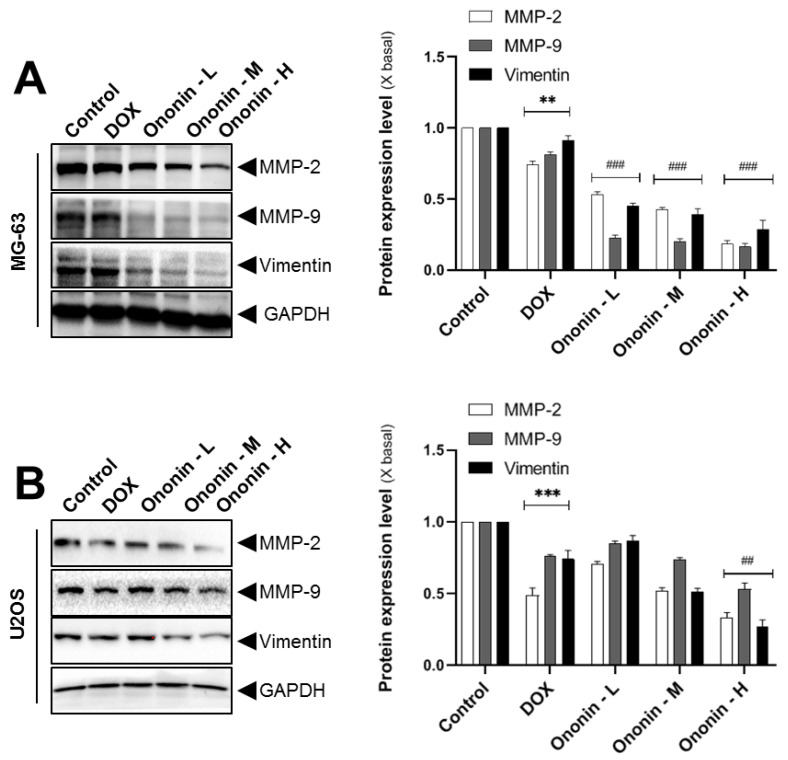
(**A**) Cultured MG-63 or (**B**) U2OS cells were incubated with DOX (1 μM) or ononin (0.1, 0.3, and 1 μM, labeled as -L, -M, and -H, respectively) for 2 days, and the target proteins were detected by Western blot (left panel). The quantification of the target protein was calculated with a densitometer (right panel). The values are expressed as the fold of changes (X basal), in Mean ± SEM, where *n* = 3. ** *p* < 0.01 and *** *p* < 0.0 1 were considered a significant result when compared to control, and ^##^ *p* < 0.01 or ^###^ *p* < 0.001 were considered significant results when compared to DOX group. Full Western blot images can be found at Appendix A.

**Figure 6 cancers-15-00758-f006:**
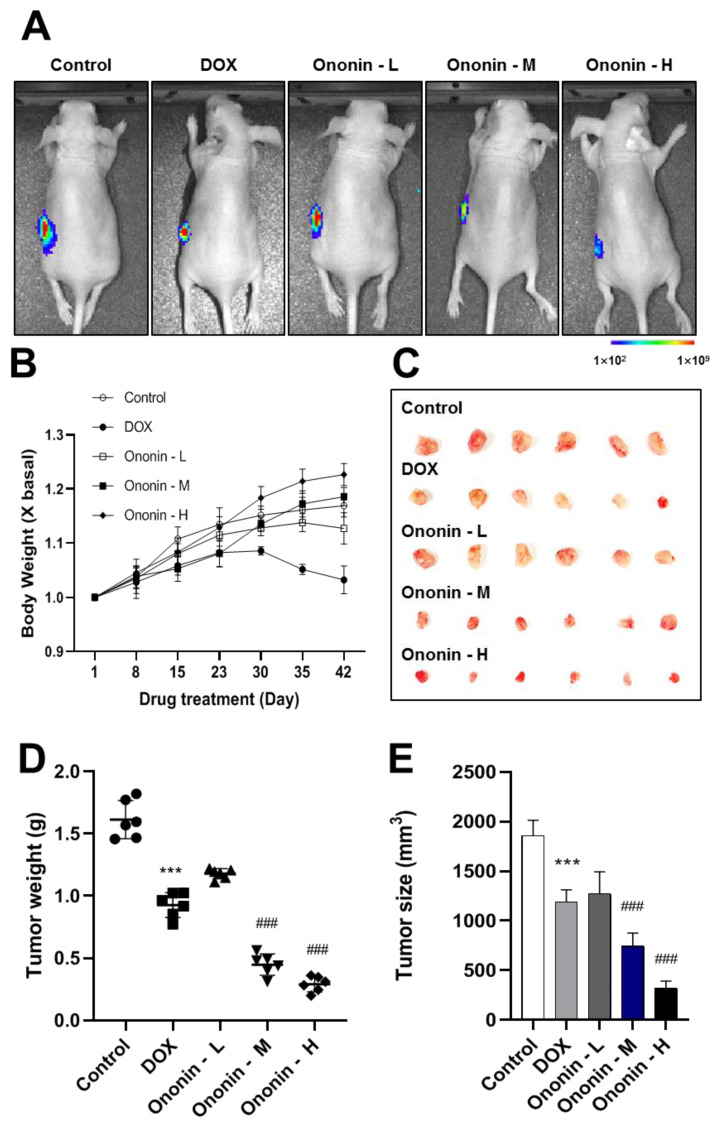
(**A**) Representative bioluminescence imaging of nude mice xenograft tumor after 42 days of treatment. The control group received PBS; the DOX group received DOX (2 mg/kg/i.p); the ononin group received different amounts of ononin (1, 3, and 10 mg/kg/i.p). (**B**) The mice’s body weight was recorded during the 42-day investigation. (**C**) On day 42, tumors were removed from mice. (**D**) After the studies, the tumor weight was calculated. (**E**) On day 42, the tumor size was calculated. The data were reported as Mean SEM, with *n* = 6. *** *p* < 0.001 was considered a significant result as compared to the control group; ^###^ *p* < 0.001 was considered a significant result as compared to the DOX group.

**Figure 7 cancers-15-00758-f007:**
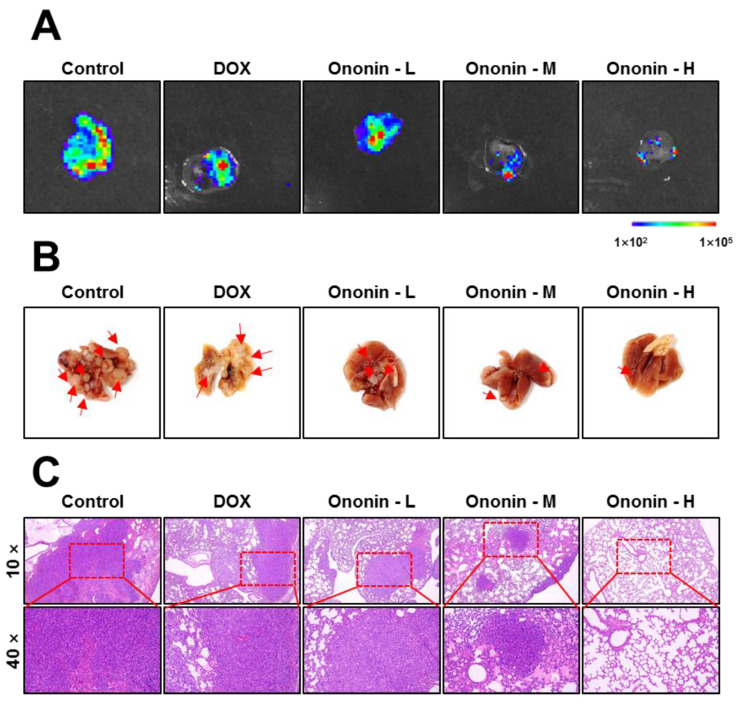
(**A**) Bioluminescence imaging of the separated lungs from each group of MG-63 xenograft nude mice. (**B**) Images of lungs from various groups and treatments. (**C**) The lung has been stained with HE. Ononin at 1, 3, and 10 mg/kg/i.p were shown as -L, -M, and -H, respectively.

**Figure 8 cancers-15-00758-f008:**
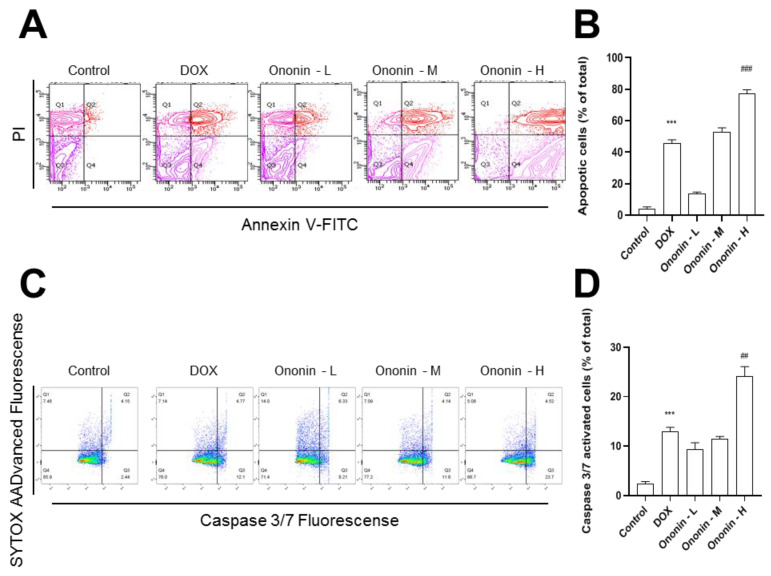
The homogenized tumor tissue was subjected to the flow cytometer for analyzing the anti-osteosarcoma cell proliferation properties. (**A**) The viable cell population was displayed in the bottom left quadrant (Q3), the early apoptotic cells in the bottom right quadrant (Q4), and the late apoptotic cells in the top right quadrant (Q4) of the dual parametric dot plots integrating annexin V-FITC and PI fluorescence (Q2). (**B**) The quantification analysis was conducted by the program Flowjo v7.6. (**C**) The dual parametric dot plots combining caspase 3/7 and SYTOX AADvanced fluorescence showed activated caspase 3/7 in the bottom right quadrant (Q4). (**D**) The quantification analysis was conducted by the program Flowjo v7.6. The values are expressed as the percentage of total cell number, *n* = 6, in Mean ± SEM. Ononin at 1, 3, and 10 mg/kg/i.p were shown as -L, -M, and -H, respectively. *** *p* < 0.001 was considered a significant result when compared to control, and ^##^
*p* < 0.01 or ^###^
*p* < 0.001 were considered significant results when compared to DOX group.

**Figure 9 cancers-15-00758-f009:**
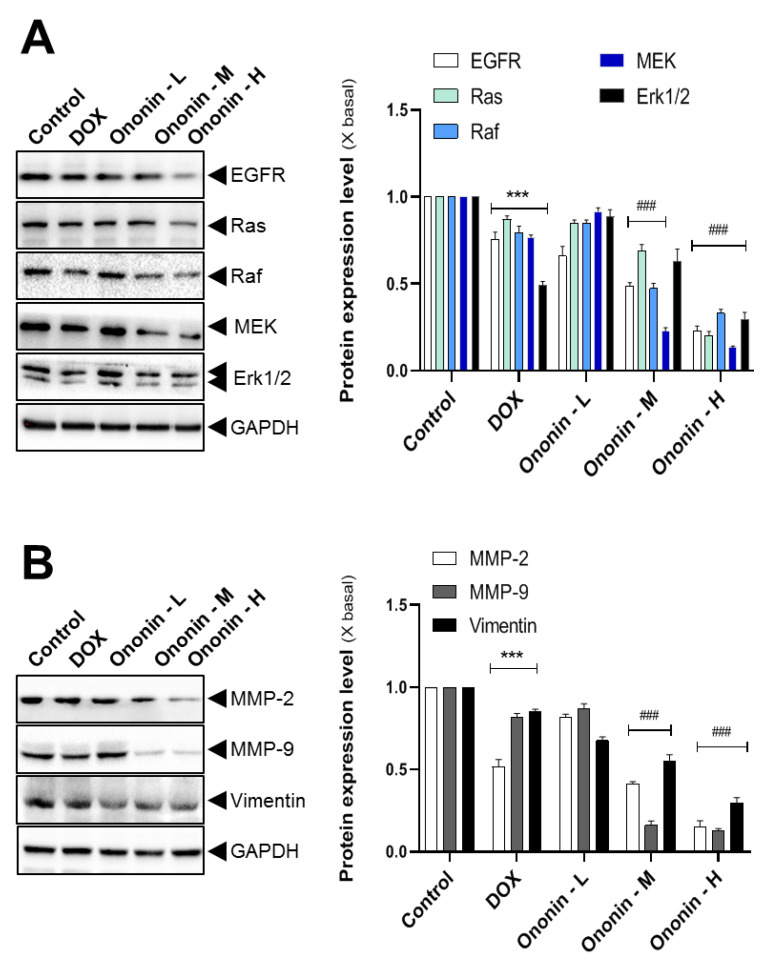
The homogenized tumor cell was subjected to SDS-PAGE for determining the target genes (**A**,**B**) translational functions. GAPDH was used as a loading control. The quantification of the target protein was calculated with a densitometer (right panel). The values are expressed as the fold of changes (X basal), in Mean ± SEM, where *n* = 6. Ononin at 1, 3, and 10 mg/kg/i.p were shown as -L, -M, and -H, respectively. *** *p* < 0.001 was considered a significant result as compared with the control, and ^###^ *p* < 0.001 was considered a significant result as compared with the DOX group. Full Western blot images can be found at Appendix A.

**Figure 10 cancers-15-00758-f010:**
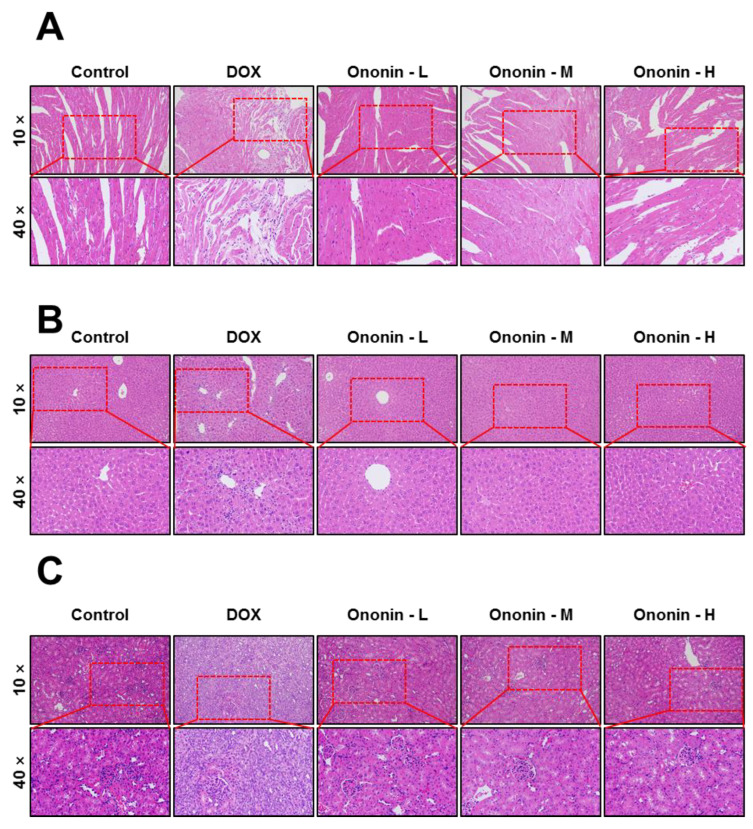
Biosafety of ononin in vivo. HE staining of (**A**) heart, (**B**) liver, and (**C**) kidney in MG-63 xenograft mice obtained from different groups. Ononin at 1, 3, and 10 mg/kg/i.p were shown as -L, -M, and -H, respectively.

**Figure 11 cancers-15-00758-f011:**
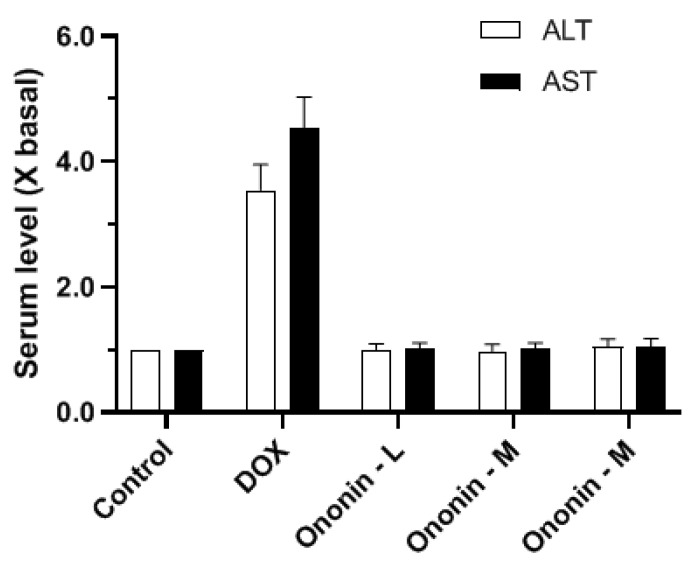
Biochemical index of animals after ononin treatment. The levels of ALT and/or AST in different groups were determined, and values are expressed as the fold of changes (X basal), in Mean ± SEM, with *n* = 6.

## Data Availability

Not applicable.

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
