# Peer review of "Anti-Invasive and Anti-Migratory Effects of Ononin on Human Osteosarcoma Cells by Limiting the MMP2/9 and EGFR-Erk1/2 Pathway"

_cancers, 2023, doi:10.3390/cancers15030758_

Round 1

Reviewer 1 Report

The Aothors should pay attention to the following issues and provide further       clarifications before the paper being accepted:

 1.    Could you explain the rational of the concentration of ononin you have used for the in vitro experiments?

2.  Why authors used DOX as positive control? 

3.   Besides the HE staining of the liver and kidney, did authors determine the ALT and AST levels of the animals?

4.    Why did authors select EGFR-MEK-Erk pathway for study?

5.    Please strengthen the English, there are too many typos within the text, professional checking and ammendment should be made.              

Author Response

The Authors should pay attention to the following issues and provide further clarifications before the paper being accepted:

Comments 1: Could you explain the rational of the concentration of ononin you have used for the in vitro experiments?

Response: Based on the MTT data in Figure 1B, we fixed the ononin concentration in the in vitro experiments. Ononin could trigger osteosarcoma cell lines (MG-63 and U2OS) death in a dose-dependent manner according to the MTT results

Comments 2: Why authors used DOX as positive control?

Response: DOX is the first-line treatment for osteosarcoma in clinical applications and is also frequently used in experiments (Maayah et al., 2018; Chen et al., 2022).

Comments 3: Besides the HE staining of the liver and kidney, did authors determine the ALT and AST levels of the animals?

Response: We have determined the ALT and AST levels in the animals, and the data were shown in Figure 11, and please find the revised file.

Comments 4: Why did authors select EGFR-MEK-Erk pathway for study?

Response: The EGFR-MEK-ERK signaling pathway is a well-studied mechanism controlling cell proliferation, metastasis, invasion, etc. (Sabbah et al., 2020). Several studies have attempted to interfere with EGFR-MEK-ERK pathway biomarkers to inhibit osteosarcoma cell proliferation and invasion (Czarnecka et al., 2020; Wilk et al., 2021).

Comments 5: Please strengthen the English, there are too many typos within the text, professional checking and ammendment should be made.  

Response: Thank you for your suggestions. We have polished the manuscript in terms of language, and all the revised parts are highlighted in red.

References

Chen Z, Yang H, Zhang Q, Hu Q, Zhao Z. Chelerythrine inhibits stemness of cancer stem-like cells of osteosarcoma and PI3K/AKT/mTOR signal. J Oncol. 2022; 6435431. 

Czarnecka AM, Synoradzki K, Firlej W, Bartnik E, Sobczuk P, Fiedorowicz M, Grieb P, Rutkowski P. Molecular biology of osteosarcoma. Cancers (Basel). 2020; 12: 2130.

Maayah ZH, Zhang T, Forrest ML, Alrushaid S, Doschak MR, Davies NM, El-Kadi AOS. DOX-Vit D, a novel doxorubicin delivery approach, inhibits human osteosarcoma cell proliferation by inducing apoptosis while inhibiting Akt and mTOR signaling pathways. Pharmaceutics. 2018; 10: 144.

Sabbah DA, Hajjo R, Sweidan K. Review on epidermal growth factor receptor (EGFR) structure, signaling pathways, interactions, and recent updates of EGFR inhibitors. Curr Top Med Chem. 2020; 20: 815-834.

Wilk SS, Zabielska-KoczywÄ…s KA. Molecular mechanisms of canine osteosarcoma metastasis. Int J Mol Sci. 2021; 22: 3639.

Reviewer 2 Report

The author showed the antimetastatic potential of ononin on human osteosarcoma cells. However, the paper needs to be modified rigorously and cannot be accepted in its current form:

Line 86 should be rephrased.

In heading 2.2 doses of compounds are not mentioned.

Line 91: the two assays can be written separately.

Line 92, 97, 108, and 177: the word medication is inappropriate for the ononin if it is not an FDA-approved drug.

Line 98: Flow cytometry is not used to label the cells with Annexin V-FITC/propidium iodide (PI) or SYTOX advanced Fluorescence. It is used to analyze the cells that are labeled with these dyes.

Line 102: incorrect unit for the volume of binding buffer. Also, a line for labeling the cells with dyes is missing.

Line 115, 247, 260, 394 and 399: the word drug is inappropriate for the ononin if it is not an FDA-approved drug.

Line 116: the line “the non-invasion cells” is incorrect.

Line 133: plural of environment and temperature should not be used here.

Figure 1-10: what refers to L, M, and H? the denotations are not mentioned in methodology or ligands with doses.

Line 205: rephrase.

The format of writing” hour” should be the same throughout the article.

Figure 6 B: according to me the bodyweight vs treatment graph should be presented using a fold change of bodyweight so that the starting point of all groups will be the same and the change can be comparable. The present graph has different starting points which is making it difficult to analyze the change.

Figure 2: Apoptosis can e presented as early, late apoptotic, or necrotic phase.

Figure 4: there is no significant gene expression change in low or high doses, check properly.

The animal result especially bio-illuminance part need to be presented properly

Author Response

Reviewer #2:

The author showed the antimetastatic potential of ononin on human osteosarcoma cells. However, the paper needs to be modified rigorously and cannot be accepted in its current form:

  1. Line 86 should be rephrased.

Response: We have rewritten the sentence.

  1. In heading 2.2 doses of compounds are not mentioned.

Response: We have included line 87 for ononin concentrations between 0 and 3 μM.

  1. Line 91: the two assays can be written separately.

Response: This has been written separately as an MTT assay and a colony formation assay.

  1. Line 92, 97, 108, and 177: the word medication is inappropriate for the ononin if it is not an FDA-approved drug.

Response: Thanks for your valuable suggestions. We have revised it accordingly.

  1. Line 98: Flow cytometry is not used to label the cells with Annexin V-FITC/propidium iodide (PI) or SYTOX advanced Fluorescence. It is used to analyze the cells that are labeled with these dyes.

Response: We have modified it according to the reviewer’s suggestions

  1. Line 102: incorrect unit for the volume of binding buffer. Also, a line for labeling the cells with dyes is missing.

Response: We have changed according to the reviewer’s suggestions.

  1. Line 115, 247, 260, 394 and 399: the word drug is inappropriate for the ononin if it is not an FDA-approved drug.

Response: We have amended the manuscript according to the reviewer’s suggestions.

  1. Line 116: the line “the non-invasion cells” is incorrect.

Response: We have corrected the inappropriate words.

  1. Line 133: plural of environment and temperature should not be used here.

Response: We have amended.

  1. Figure 1-10: what refers to L, M, and H? the denotations are not mentioned in methodology or ligands with doses.

Response: We have included the detailed information on each figure legend.

  1. Line 205: rephrase.

Response: We have revised it.

  1. The format of writing” hour” should be the same throughout the article.

Response: We have formatted the text. 

  1. Figure 6 B: according to me the bodyweight vs treatment graph should be presented using a fold change of bodyweight so that the starting point of all groups will be the same and the change can be comparable. The present graph has different starting points which is making it difficult to analyze the change.

Response: We have changed Figure 6B according to the reviewer’s suggestions.

  1. Figure 2: Apoptosis can be presented as early, late apoptotic, or necrotic phase.

Response: We calculated the percentage of live cells in different treatments to provide a better understanding

  1. Figure 4: there is no significant gene expression change in low or high doses, check properly.

Response: We have included the statistical symbol in Figure 4.

  1. The animal result especially bio-illuminance part need to be presented properly

Response: It has been included in the results part.

Reviewer 3 Report

The author has presented a well study of anti-invasive and anti-migratory effects of ononin.

1.     How is the effect of shear stress and ononin on the survival and proliferation of circulating tumour cells?

Please refer to the:

·       https://aacrjournals.org/cancerres/article/77/13_Supplement/2325/617808/Abstract-2325-Destruction-of-circulating-tumor

·       Applications of microfluidics and organ-on-a-chip in cancer research." Biosensors 12, no. 7 (2022): 459.

2.     Explain the dose dependent of ononin treatment of metastatic breast cancer cells like MDA-MB-231 as well as non-metastatic cells like MCF7.

Please refer to the effect of Shear stress and doxorubicin in following article and examine similarly with the ononin.:

·        Breast Cancer Research and Treatment 172, no. 2 (2018): 297-312.

3.     Please test the effect of ononin against the highly metastatic cells line as given in Regmi, S., Fu, A., & Luo, K. Q. (2017). Scientific reports7(1), 1-12.

4.     Please make in vitro italic everywhere in the text.

5.     After 2 h incubation in MTT assay, people usually keep 100 μl of solubilization solution containing 10% SDS and 0.1% HCl to each well. How much DMSO was added after 2 hours of incubation?

6.     In your figure 1, can you add the data of MG-63 and U2OS cells growth in an absence of DOX and ononin? We need this to see the cell morphology is good during the experiment.

7.     What do you mean by ononin-L, M and H? The purpose of writing is to make reader clear, please do not assume that reader will know it on its own. Therefore, I request to make the test as simple as possible so every reader can understand.

8.     Why is the data shown for 48 hours only in figure 1? Please elaborate it for 24 hours as well as 72 hours condition.

9.     In your figure 1, C you showed you showed after 48 hours of medication treatment, for another 7 days. We would like to see the response for 1 day, 2 days, 3 days, 5 days in between, please address this concern.

10.  In your figure B, can you show similar apoptosis cells by DOX and Breast Cancer Research and Treatment 172, no. 2 (2018): 297-312.

11.  Please run a western blot to detect caspase-3 (to see the consistent result as Figure 2D).

12.  In Figure 3, the representative figure to show Ononin suppression cell migration and invasion does not look so difference. Please quantify the figure 3 and tell in terms of percentage about the cell migration and invasion.

13.  How was the protein expression calculated in case of EGFR, Ras, Raf, MEK, and Erk1/2 in figure 4?

14.  The expression of MMP2/9 in was tested in two difference cell lines: human osteosarcoma MG-231 63 and U2OS cells. Can you provide one more cell line for consistency in figure 5? What about like MDA-MB-231 / MCF-7 cell lines?

15.  The author mentioned that “The animals were sacrificed after receiving the drug treatment for 42 days.” Why was these 42 days of timepoint chosen in the study?

16.  The author showed that Ononin decreases xenograft mice tumour size in 42 days in figure 6. Will the tumour size go completely zero if the time is doubled, tripled? Please show the experimental data for 84 days (double) or 126 days (tripled) too.

17.  How many mice was taken in Figure 8 to show Ononin declines osteosarcoma cells proliferation in vivo?

18.  Please add the rationale of conducting the biosafety of ononin in vivo in discussion part.

19.  What are the further experimental directions in the field? Please discuss in conclusion section.

20.  Please provide the key novelty and significance of the study in the introduction section.

The author should address above all the concern before I can recommend the manuscript for acceptance. For now, I would like to reject the manuscript.

Author Response

Reviewer #3:

  1. How is the effect of shear stress and ononin on the survival and proliferation of circulating tumour cells? Please refer to the: https://aacrjournals.org/cancerres/article/77/13_Supplement/2325/617808/Abstract-2325-Destruction-of-circulating-tumor Applications of microfluidics and organ-on-a-chip in cancer research." Biosensors 12, no. 7 (2022): 459.

Response: The main objective of this manuscript was to examine the detailed working mechanism of ononin on human osteosarcoma cell lines in vitro and in vivo. The cell viability was determined by the MTT assay, and the apoptotic cell was analyzed by the flow cytometer.

  1. Explain the dose dependent of ononin treatment of metastatic breast cancer cells like MDA-MB-231 as well as non-metastatic cells like MCF7. Please refer to the effect of Shear stress and doxorubicin in following article and examine similarly with the ononin. Breast Cancer Research and Treatment 172, no. 2 (2018): 297-312.

Response: It is the purpose of the study to determine the effects of anti-osteosarcoma cells on proliferating osteosarcoma cells and resistance to metastatic spread rather than the mechanism of breast cancer. 

  1. Please test the effect of ononin against the highly metastatic cells line as given in Regmi, S., Fu, A., & Luo, K. Q. (2017). Scientific reports, 7(1), 1-12.

Response: A type of highly metastatic cancer type commonly found in young people, osteosarcoma, is often diagnosed with lung metastasis clinically (Yang et al., 2020; Sheng et al., 2021). It is the purpose of the study to determine the effects of anti-osteosarcoma cells on proliferating osteosarcoma cells

  1. Please make in vitro italic everywhere in the text.

Response: We have amended according to the reviewer’s suggestion.

  1. After 2 h incubation in MTT assay, people usually keep 100 μl of solubilization solution containing 10% SDS and 0.1% HCl to each well. How much DMSO was added after 2 hours of incubation?

Response: The MTT solution was discarded and added DMSO solution.

  1. In your figure 1, can you add the data of MG-63 and U2OS cells growth in an absence of DOX and ononin? We need this to see the cell morphology is good during the experiment.

Response: Both MG-63 and U2OS were obtained from ATCC, and the passage number used for the experiment were all less than 8, the morphology was determined before the experiment. 

  1. What do you mean by ononin-L, M and H? The purpose of writing is to make reader clear, please do not assume that reader will know it on its own. Therefore, I request to make the test as simple as possible so every reader can understand.

Response: Ononin-L, -M, and -H mean low dosage, medium dosage, and high dosage. We have highlighted the legend of the figure.

  1. Why is the data shown for 48 hours only in figure 1? Please elaborate it for 24 hours as well as 72 hours condition.

Response: Before doing the full experiment, we optimized treatment duration, and we found the condition of the cells and the results are much better after receiving 48 hours of treatment

  1. In your figure 1, C you showed you showed after 48 hours of medication treatment, for another 7 days. We would like to see the response for 1 day, 2 days, 3 days, 5 days in between, please address this concern.

Response: In the colony formation assay, we treated the cells for 48 hours with ononin and then changed the medium for another 7 days.  

  1. In your figure B, can you show similar apoptosis cells by DOX and Breast Cancer Research and Treatment 172, no. 2 (2018): 297-312.

Response: In the MTT assay, we treated the osteosarcoma cells with ononin (0-3 μM) for 48 hours (Figure 2B) and DOX (Figure 2A).

  1. Please run a western blot to detect caspase-3 (to see the consistent result as Figure 2D).

Response: MTT, colony formation, and flow cytometry are three methodologies we used to reflect the cell proliferation condition. All the data indicated that ononin was able to suppress osteosarcoma viability.

  1. In Figure 3, the representative figure to show Ononin suppression cell migration and invasion does not look so difference. Please quantify the figure 3 and tell in terms of percentage about the cell migration and invasion.

Response: As stated in the methodology and figure legend, no Matrigel-coated transwells were used to test the effect of cell migration. Matrigel-coated transwells were used to analyze invasion functions. 

  1. How was the protein expression calculated in case of EGFR, Ras, Raf, MEK, and Erk1/2 in figure 4?

Response: The protein translation levels were calculated based on the western band intensity and Image J software.

  1. The expression of MMP2/9 in was tested in two difference cell lines: human osteosarcoma MG-231 63 and U2OS cells. Can you provide one more cell line for consistency in figure 5? What about like MDA-MB-231 / MCF-7 cell lines?

Response: This study clarifies Ononin's treatment of osteosarcoma in vitro and in vivo. 

  1. The author mentioned that “The animals were sacrificed after receiving the drug treatment for 42 days.” Why was these 42 days of timepoint chosen in the study?

Response: Considering that the tumor burden in the control group had already become a burden on the mice, the animals must be sacrificed in accordance with ethical policy.

  1. The author showed that Ononin decreases xenograft mice tumour size in 42 days in figure 6. Will the tumour size go completely zero if the time is doubled, tripled? Please show the experimental data for 84 days (double) or 126 days (tripled) too.

Response: On day 42, the nude mice were already affected by tumor burden, and in accordance with ethical policy, the animals were to be sacrificed.

  1. How many mice was taken in Figure 8 to show Ononin declines osteosarcoma cells proliferation in vivo?

Response: Each group has 6 animals, and n = 6.

  1. Please add the rationale of conducting the biosafety of ononin in vivo in discussion part.

Our reply: Osteosarcoma patients are currently treated with chemotherapy, which has severe side effects. We still need to consider the safety of ononin before performing further clinical studies to determine if it is effective in treating osteosarcoma.

  1. What are the further experimental directions in the field? Please discuss in conclusion section.

Response: It has been included in the conclusion

  1. Please provide the key novelty and significance of the study in the introduction section.

Response: It has been included in the introduction.

Reference:

  1. Yang C, Tian Y, Zhao F, Chen Z, Su P, Li Y, Qian A. Bone microenvironment and osteosarcoma metastasis. Int J Mol Sci. 2020; 21: 6985
  2. Sheng G, Gao Y, Yang Y, Wu H. Osteosarcoma and metastasis. Front Oncol. 2021; 11: 780264.

Round 2

Reviewer 2 Report

All the requested changes are well answered by the author

Author Response

We would like to thank you for your time and effort in reviewing our manuscript, and for your valuable suggestion to improve the quality of our manuscript. The minor changes have been marked in red color in the new version; we have done our best to improve the quality of the paper.

Reviewer 3 Report

In the colony formation assay, after the cells was treated for 48 hours with ononin, why the medium was changed for another 7 days? Why the time period of 1 day, 2 days, 3 days, 5 days in between is not considered? Please address this concern.

In addition to MTT, colony formation, and flow cytometry methodologies to reflect the cell proliferation condition, I am curious to see a western blot to detect caspase -3 too for apotosis. Do you use the necrotic cells too?

Author Response

Comments: In the colony formation assay, after the cells was treated for 48 hours with ononin, why the medium was changed for another 7 days? Why the time period of 1 day, 2 days, 3 days, 5 days in between is not considered? Please address this concern.

Response: The colony formation assay aims to detect cell proliferation efficiencies. After drug treatment for 2 days, and the medium was changed by fresh medium every 48 hours

Comments: In addition to MTT, colony formation, and flow cytometry methodologies to reflect the cell proliferation condition, I am curious to see a western blot to detect caspase -3 too for apotosis. Do you use the necrotic cells too?

Response: Caspase-3 is a biomarker for apoptosis; however, it is not able to reflect the cells undergoing early or late apoptotic status. PI/Annexin-FITC coupled with flow cytometry is able to indicate the exact cell status. Here, we used PI/Annexin-FITC equipped with flow cytometry to address this issue in Figure 2A.

Round 3

Reviewer 3 Report

The authors did not answer well.